# Estimation of the case fatality rate based on stratification for the COVID-19 outbreak

**Byungwon Kim, Seonghong Kim, Woncheol Jang●\*, Sungkyu Jung, Johan Lim**

Department of Statistics, Seoul National University, Seoul, Republic of Korea

\* wcjang@snu.ac.kr

## Abstract

This work is motivated by the recent worldwide pandemic of the novel coronavirus disease (COVID-19). When an epidemiological disease is prevalent, estimating the case fatality rate, the proportion of deaths out of the total cases, accurately and quickly is important as the case fatality rate is one of the crucial indicators of the risk of a disease. In this work, we propose an alternative estimator of the case fatality rate that provides more accurate estimate during an outbreak by reducing the downward bias (underestimation) of the naive CFR, the proportion of deaths out of confirmed cases at each time point, which is the most commonly used estimator due to the simplicity. The proposed estimator is designed to achieve the availability of real-time update by using the commonly reported quantities, the numbers of confirmed, cured, deceased cases, in the computation. To enhance the accuracy, the proposed estimator adapts a stratification, which allows the estimator to use information from heterogeneous strata separately. By the COVID-19 cases of China, South Korea and the United States, we numerically show the proposed stratification-based estimator plays a role of providing an early warning about the severity of a epidemiological disease that estimates the final case fatality rate accurately and shows faster convergence to the final case fatality rate.

**Data Availability Statement:** The data underlying the results presented in the study are available from https://github.com/sungkyujung/covid19cfr and all R functions and example codes used for the calculation of the proposed CFR estimates are also

## Introduction

The outbreak of the novel coronavirus disease (COVID-19) epidemic in Wuhan City, Hubei Province, China on December 8, 2019 has sharply grown as a global pandemic. The reported total number of confirmed cases worldwide has exceeded 2.4 million (2,418,720), including 166,276 deaths as of April 20, 2020 [1]. In the early stage of the epidemic, the most cases reported outside Hubei had a history of travel to Wuhan which had effects on the growth of the disease in a few Asian countries, but the impact of recent secondary infections has been wider and faster in the world. To reduce the spread of epidemics, a few policies such as wearing a mask and social distancing are suggested in many countries. See [2] for a research on the effects of social distancing which supports the policy. However, since these policies are only for prevention, they are not enough to overcome the pandemic.

The case fatality rate (CFR) is one of the important epidemiological measures of disease severity, and is the proportion of deaths among the cases of the disease. As the importance of

posted. Daily updated estimates of CFRs (1. Age-group stratified CFRs in South Korea, 2. State stratified CFRs in the United States, 3. Province stratified CFRs in China, and 4. Country stratified global CFRs) are posted online,'https://sungkyujung.github.io/covid19cfr/'.

**Funding:** W.J. and B.K. were supported by the National Research Foundation of Korea (NRF) grant and a grant of the Korea Health Technology R&D Project through the Korea Health Industry Development Institute (KHIDI) funded by the Korea government(MSIT) and the Ministry of Health & Welfare, Republic of Korea (No. 2017R1A2B2012816, HI19C0378). S.J. and B.K. were supported by the National Research Foundation of Korea (NRF) grant funded by the Korea government (MSIT) (No. 2019R1A2C2002256).

**Competing interests:** The authors have declared that no competing interests exist.

quick and accurate estimation of the CFR during a disease outbreak is emphasized by many researchers [3–8], the world health organization (WHO) and many countries report the naive CFR daily basis. The naive CFR is the proportion of the cumulative number of deaths out of the cumulative number of confirmed cases at the time. It is well known that the naive CFR has a tendency of underestimation due to not counting the quarantined patients at the time, some of which will eventually die.

Many other estimators of the CFR have been proposed to reduce the bias of the naive CFR. For example [9–14], proposed to adjust the number of confirmed cases in the denominator by multiplying a factor [15]. Proposed to adjust the numerator by adding the expected fatalities of quarantined patients, the expected fatalities are computed based on the duration of hospitalization.

The reason for using the naive CFR by the WHO and many other countries instead of those proposed alternatives [9–15] is that the naive CFR requires minimum information for computation and eventually converges to the final CFR, the proportion of overall deaths out of the total cases when the epidemic is over. During the pandemic of a disease, it is practically infeasible to record all individuals' information accurately and to report it to the public, but most of those proposed estimators require extra personal information. In particular, the estimator of [15] requires daily updates of individuals' statuses from confirmed dates, which is unavailable. For example, in South Korea, since the COVID-19 epidemic started to grow sharply in Daegu on Feb 18, the government has stopped reporting the individual records of infected patients, such as the number of contacts with people, the list of public places visited and the status changes of a patient.

In this paper, we propose an alternative approach for the real-time estimation of CFR. It is designed to reduce the bias of the naive CFR, by accounting for the expected future fatalities as done in [15]. To accommodate the real-time update, the proposed model requires only the cumulative numbers of confirmed, deceased and cured patients at the time. While using these simple quantities, to properly adapt characteristics of different groups or partitions in the population, we use a stratification, which estimates the expected number of deaths among the quarantined patients in each stratum separately and then combines those estimated counts to achieve an asymptotic consistency to the final CFR.

The proposed stratification-based model is applied to the cases of China, South Korea, and the United States. For the case of South Korea, the age-group is used for the stratification. For the cases of China and the United States, provinces and states are used for the stratification, respectively. From these examples of application, we show that the proposed model suitably reduces the bias of the naive CFR and converges to the final CFR. The benefit of stratification is greater when we have more heterogeneous strata, e.g. the age-group in South Korea, which results in faster convergence to the final CFR than the naive CFR. As a result, by providing an estimate of the case fatality rate that is less biased and converges to the final CFR faster than the naive CFR, the proposed stratification-based model plays a role of providing an early warning of the severity of a disease.

## Methods

We now provide the proposed model for the real-time estimation of the case fatality rate of a disease. Suppose $N(t)$, $C(t)$, $D(t)$, $U(t)$ be the cumulative numbers of the infected, cured, deceased, and quarantined patients up to time $t$. Here, the quarantined patients include patients in the hospital and self-quarantined patients. The final turnout of the quarantined patients is unknown but they will be eventually cured or deceased. At each $t$, $N(t) = C(t) + D(t) + U(t)$. We propose an estimator of the final CFR, called a modified naive CFR, up to

time $t$ as

$$\widehat{\text{FR}}(t) = \frac{D(t) + E_u^d(t)}{N(t)}, \tag{1}$$

where $E_u^d(t)$ denotes the estimated count of deaths among the quarantined patients $(U(t))$ up to time $t$.

The proposed model is designed under an assumption that there exist heterogeneous strata in which the numbers of confirmed, cured, deceased and quarantined patients are available for each stratum. These stratum-wise quantities are used in the calculation of $E_u^d(t)$. Let $N_l(t)$, $C_l(t)$, $D_l(t)$, $U_l(t)$ be the cumulative counts of the infected, cured, deceased and quarantined patients for the stratum $l = 1, \ldots, L$. When there is no stratum-wise information, $E_u^d(t)$ is estimated by letting $L = 1$. We now assume, for each stratum $l$, that given $N_l(t)$,

$$(C_l(t), D_l(t), U_l(t)) \sim \text{Multinomial}(N_l(t), (p_c(l, t), p_d(l, t), p_u(l, t))),$$

and $(C_l(t), D_l(t), U_l(t))$ and $(C_\ell(t), D_\ell(t), U_\ell(t))$ are independent for $l \neq \ell$. Here, $(p_c(l, t), p_d(l, t), p_u(l, t))$ is the time-varying proportion of the cured, deceased and quarantined patients for stratum $l$. We also assume $p_u(l, t) \rightarrow 0$ as $t \rightarrow \infty$. Recall, at the end of an epidemic, all confirmed patients are determined to be either cured or deceased.

The proposed estimator for the $l$-th stratum case fatality rate is now defined by

$$\widehat{\text{FR}}_l(t) = \frac{D_l(t) + E_u^d(l, t)}{N_l(t)} = \frac{D_l(t)}{N_l(t)}\left(1 + \frac{U_l(t)}{N_l(t)}\right). \tag{2}$$

Here, $E_u^d(l, t)$ denotes the estimated count of deaths among the quarantined patients in $l$-th stratum, and $E_u^d(t)$, in (1) is then defined as the sum of $E_u^d(l, t)$'s, which leads to the proposed estimator of the case fatality rate (1) as

$$\widehat{\text{FR}}(t) = \frac{D(t) + \sum_{l=1}^{L} E_u^d(l, t)}{N(t)} = \sum \frac{N_l(t)}{N(t)}\widehat{\text{FR}}_l(t). \tag{3}$$

Note that the proposed estimator of the final CFR (3) is a weighted average of the estimator of the stratum-wise CFR (2).

During an epidemic, the proposed estimators (2) and (3) prevent underestimation of the case fatality rate by adding the estimated counts of deaths among the quarantined patients to the naive CFR. In addition to the bias correction, the proposed estimator has consistency such that it converges to the final CFR as the outbreak of a disease coming over. To see the convergence, let $p_d(l, \infty)$ and $p_d(\infty)$ be the final case fatality rates for a stratum $l, l = 1, \ldots, L$, and for the population, respectively. Here, we use the infinity symbol, $\infty$, to denote the time-point of the end of epidemic. Though duration of the epidemic is practically finite, but it is unknown during an epidemic, which makes reasonable to use the infinity symbol to indicate an unknown time-point in the future. Recall that the final case fatality rate of a disease is the overall proportion of deaths out of total cases at the end of epidemic, which means there is no more quarantined patients not yet determined to be cured or deceased. This satisfies our assumption, $p_u(l, t) \rightarrow 0$ as $t \rightarrow \infty$, which leads to $E_u^d(l, \infty) \rightarrow 0$ and $E_u^d(\infty) \rightarrow 0$. Hence, $\widehat{\text{FR}}_l(t) \rightarrow p_d(l, \infty)$ and $\widehat{\text{FR}}(t) \rightarrow p_d(\infty)$ as $t \rightarrow \infty$.

Additional observations for the proposed estimators (2) and (3) are given in the S1 Appendix. In particular, Theorem 1 (ii) and (iii) provide the variances of (2) and (3) which can be used for approximate confidence intervals.

## Daily Case Fatality Rate in China

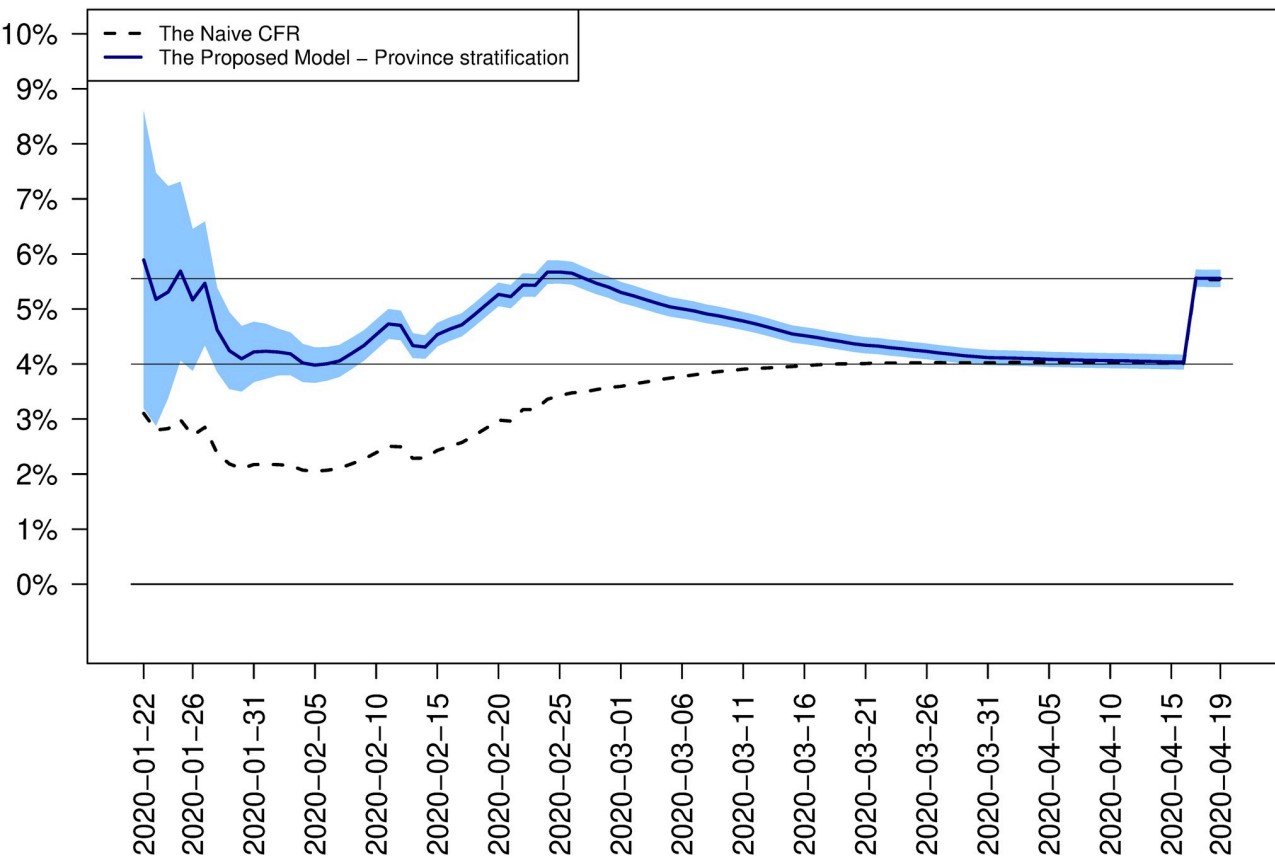

**Fig 1. Daily estimates of CFRs in China.** Two CFR estimates in China, the daily updated naive CFRs (dotted line) and the proposed province stratified estimates of CFRs (blue solid line) are depicted together. For our proposed estimates, daily basis 95% confidence intervals (shaded area) are also added.

## Results

### Province stratified case fatality rates in China

First of all, we applied the proposed model to the estimation of the final case fatality rate in China. Because the COVID-19 disease was first emerged in Wuhan City China and the first wave of the COVID-19 pandemic in China was almost over, it was meaningful to apply our model to the China case to see the entire flow of the estimated case fatality rates. The data used for the analysis here is available on [16].

The proposed estimator of the final case fatality rate is computed daily basis. Since there were no demographic factors such as age and gender, which the factor-wise information including the numbers of confirmed, deceased and cured patients in each factor group was available, we used the provinces in China for the stratification. Fig 1 shows the estimated CFR. One can see an weird jump in both the naive CFR and the proposed estimate on Apr 17, which comes from the sudden addition of 1,290 deceased cases in Wuhan announced by the government of Wuhan City, see [17, 18] for details. Due to the late report of the missed fatalities, the data published by the government of Wuhan City and followed statistics including the

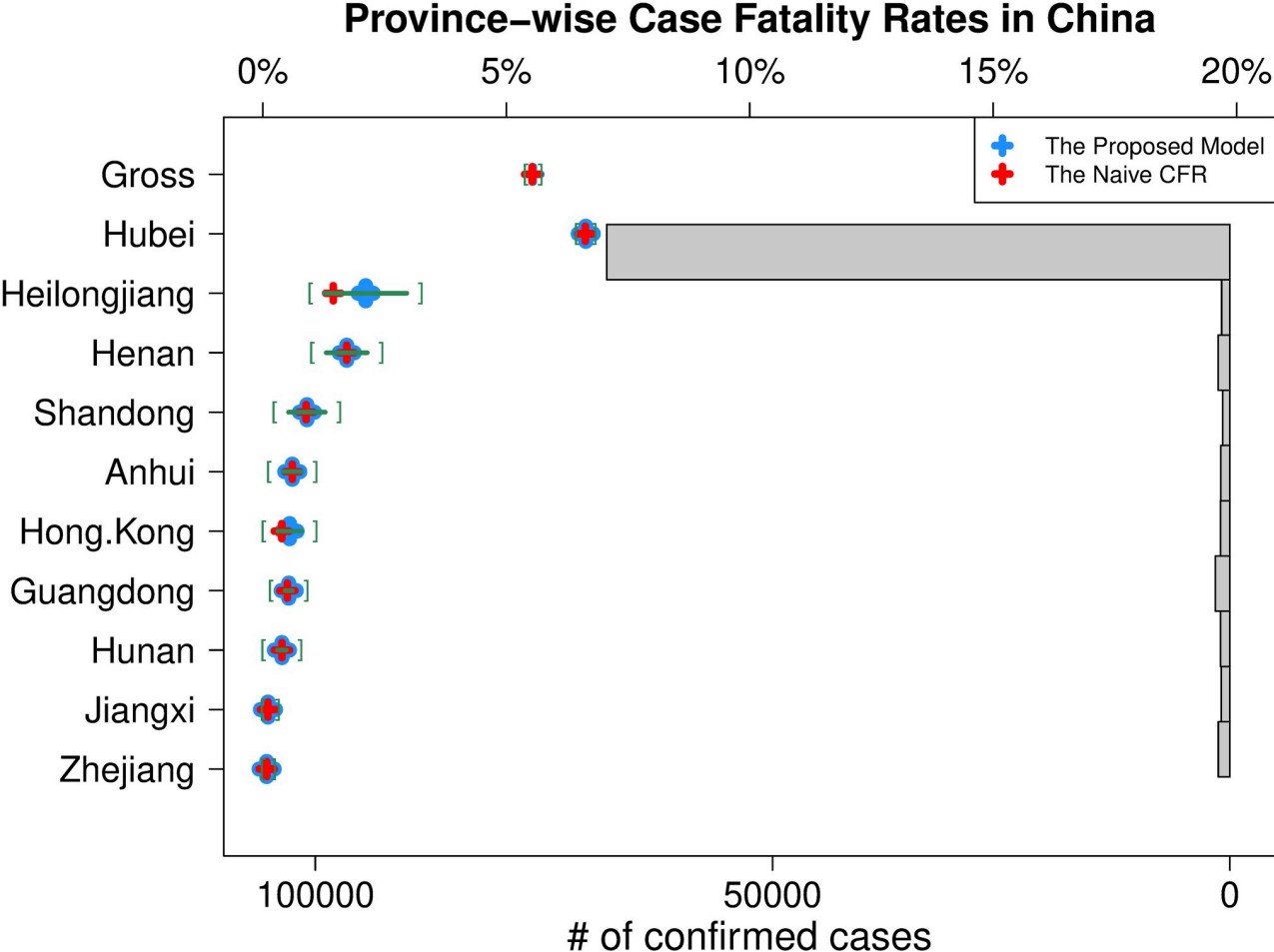

**Fig 2. Latest province-wise CFRs in China.** For top 10 provinces in the numbers of confirmed cases, the latest proposed estimate of CFR (blue plus marker) and its 95% confidence interval (green square brackets) for each province is depicted with the histogram of confirmed cases. The latest naive CFR (red plus marker) calculated for each province separately is also added to show the difference.

estimates of the CFR are questionable, but still there is room for interpretation of the flow of our proposed estimates.

When assumed there is no missed fatalities, we see that the proposed estimate computed daily basis has steadily given a warning that the COVID-19 disease is much more danger than what the naive CFR says during the outbreak. In particular, on Jan 31 from which the number of confirmed cases has grown exponentially, our estimate of CFR was 4.22% (3.67%, 4.77%) that is close to the final CFR while the naive CFR was only 2.17% (1.88%, 2.46%). According to Fig 1, the proposed estimate hit the second peak on Feb 25, which is 5.67% (5.46%, 5.88%), and then gradually decrease towards 4%. However, if the missed deceased cases were accurately reported from the first, we expect that such a decrease would not happen, which leads to a faster convergence to the current final CFR, 5.55%.

Fig 2 shows the latest province-wise estimates of CFRs with 95% confidence intervals for top 10 provinces in the numbers of confirmed cases. Almost confirmed cases (about 81%) have been reported in Hubei province whose naive CFR is 6.62% (6.43%, 6.81%) and our model estimate is 6.63% (6.45%, 6.82%) on Apr 19, 2020, while the CFR estimates of other provinces are so much smaller than the estimated CFR.

## Daily Case Fatality Rate in South Korea

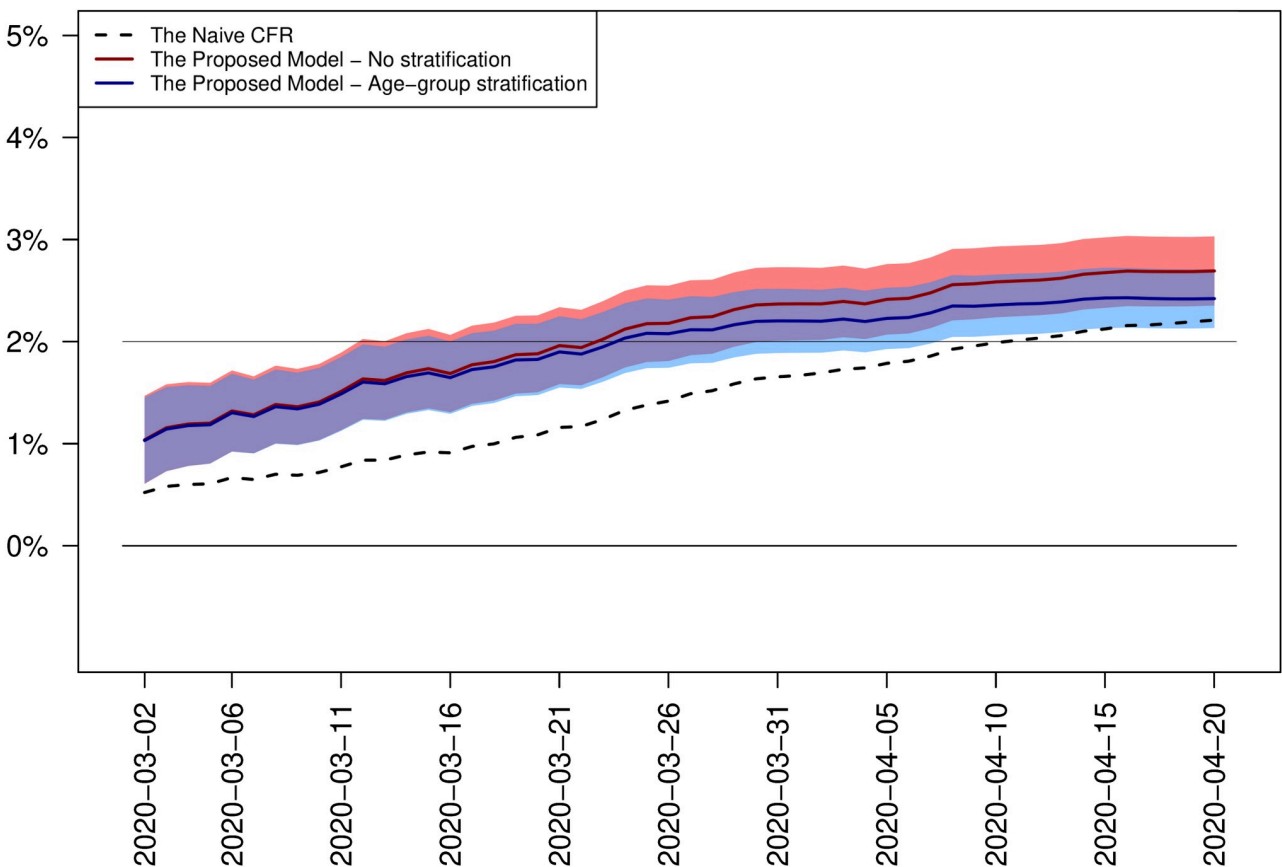

**Fig 3. Daily estimates of CFRs in South Korea.** The daily updated naive CFRs (dotted line), the estimates of CFRs using our model without stratification (red solid line), and the estimates of CFRs using our model with age-group stratification (blue solid line) are depicted together. For our model estimates, both without and with age-group stratification, daily basis 95% confidence intervals (shaded areas) are also added.

### Age-group stratified case fatality rates in South Korea

Next, we applied the proposed model to the estimation of the case fatality rate of the COVID-19 epidemic in South Korea. We used age-groups for stratification. The first confirmed case in South Korea was reported on Jan 20, and the first death was reported on Feb 20, 2020. We use the data from Mar 2 as the Korea center for disease control and prevention (KCDC) and the ministry of health and welfare (MOHW) stopped data publication from Feb 18 to Mar 1, and resumed on March 2. To use age-group information, the results including the below figures are obtained using the data reported by MOHW from Mar 2 to Apr 20, 2020 [19]. The data is also available on [20].

As we did in the case of China, our estimator of the final CFR is computed on a daily basis, and we see that it consistently outperforms the naive CFR at each time. To confirm this, we have plotted the proposed estimates of CFR (with ± 2 standard error) and the naive CFR over time. Fig 3 shows the daily estimated CFRs using the naive CFR, the proposed estimator with and without age-group stratification. From the figure, we first see that there has been no dramatic change unlike the case of China, where we see fluctuations in late January, over time in

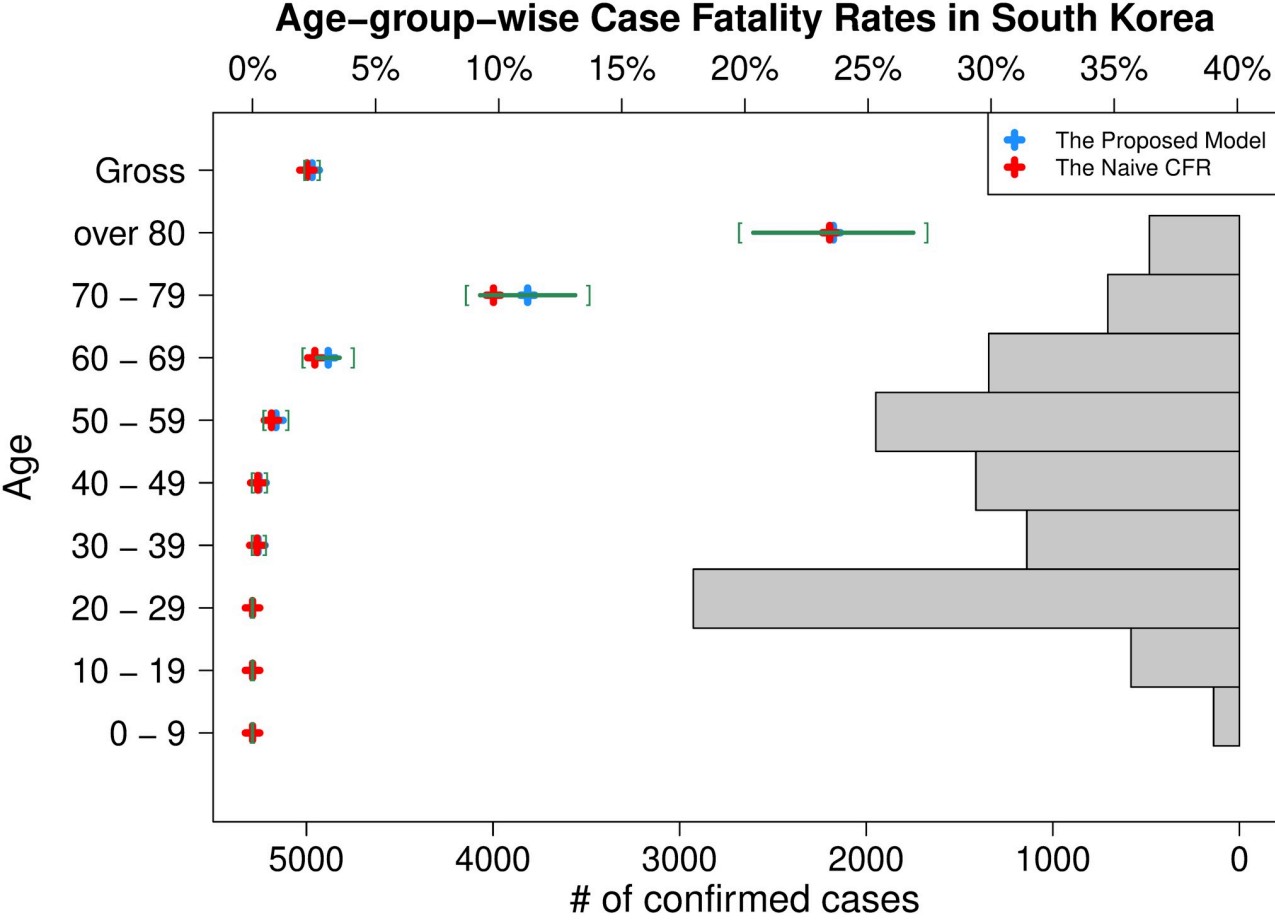

**Fig 4. Latest age-group-wise CFRs in South Korea.** The latest estimate of CFR (blue plus marker) and its 95% confidence interval (green square brackets) for each age-group using the proposed age-group stratification model is depicted with the histogram of confirmed cases. The latest naive CFR (red plus marker) calculated for each age-group separately is also added to show the difference.

daily estimated CFRs of all three estimates but those CFRs have steadily been increasing. Second, we find that the naive CFR is always outside the 95% confidence intervals of the proposed estimates (both with and without age-group stratification), showing clear underestimation. Third, from around Mar 25, the rate of increase in the proposed age-group stratified CFR estimate has started to be slower than other two, which can be understood as a signal of fastest convergence to the final CFR. This also shows the stratification can reduce an upward bias which might happen when our proposed model is used without any stratification. Last, on Apr 20, our estimate with age-group stratification was 2.42% with 95% confidence interval (2.13%, 2.71%) while the naive CFR was 2.21% (1.93%, 2.49%). The naive CFR has been inside the confidence interval of the proposed age-group stratified estimate recently, which means the convergence to the final case fatality rate.

To see a stratification effect, how heterogeneous strata affect on the case fatality rate, we need to check the stratum-wise estimates as well. According to Fig 4, which shows the latest age-group-wise estimates of CFRs with 95% confidence intervals, the age clearly has a positive relationship with the case fatality rate. There has been no deceased patient in the young-age groups (less than 30 years) while almost deceased cases were reported in the old-age groups (more than 50 years). The figure shows that the estimated CFR for the group of 80s or over is

23.59% (19.79%, 27.39%) and for the group of 70s is 11.18% (8.69%, 13.66%), both are so much higher than the estimated CFR. Based on the result, we notice that the young age groups, especially the group of 20s, are evidently dropping the CFR by taking large proportions in confirmed cases (largest in 20s) with no deceased cases. In fact, this phenomenon that the group of 20s takes the largest proportion in the number of confirmed cases and lowers the CFR in a significant degree is highly related with a religious cult, Shincheonji, in South Korea, of which a member played a key role for the spread of the COVID-19 infections in Daegu. The Korean government decided to diagnose all members of the cult, including people without a symptom of the disease. By the decision, the group of 20s became the largest age-group in the confirmed cases, but most of them had no symptom, which affected on lowering the CFR.

### State stratified case fatality rates in the United States

The last example is the case of the United States. The recent growth of the COVID-19 pandemic in the United States is much severer than any other country that the numbers of confirmed and deceased cases have exceeded 700 thousand and 37 thousand, respectively, as of April 20, 2020, both are now the most cases in the world. The data used for the analysis here is available on [21].

Fig 5 shows the estimated daily case fatality rates in the United States. We find that the flow of CFR estimates in the United States has just passed the first wave, the first sharp increase and decrease of CFRs in the earliest stage of pandemic, which does not give a clue for the final case fatality rate. The figure shows that, after the first wave, both the naive CFR and the proposed estimate with state stratification are still increasing and the gap between these two CFRs is getting wider. Based on the cases of South Korea and China above, it may take longer than expected to see the decrease of the daily CFRs and the final convergence. And the state-wise estimates of CFRs, which show significant differences between the proposed estimate and the naive CFR in all states as depicted in Fig 6, also support this expectation.

As the histogram in Fig 6 shows, We know that the number of confirmed cases in the New York state overwhelms other states in the United States, but the estimated CFR, 10.47% (10.30%, 10.63%), takes the second place, while the Michigan state is ranked at the first with the estimated CFR, 13.79% (13.27%, 14.31%), which is somewhat out of expectation. As analyzed in [22], the reason that the estimated CFR is the highest in the Michigan state might come from the low rate of COVID-19 test, 795 tests taken per 100 thousand people in the Michigan state that is ranked 27th in the United States, but further analysis such as demographic factors stratified estimation of the case fatality rate can be helpful for figuring out the reason.

### Discussion

In this paper, we proposed a modification of the naive CFR that allows to estimate the case fatality rate of a disease in real-time basis. The proposed estimator is designed to reduce the downward bias (underestimation) of the naive CFR during an outbreak of a disease, especially in the early stage of an epidemic, and to converge to the final case fatality rate. In particular, by adapting a stratification, the proposed estimator is made to combine information from heterogeneous strata appropriately. By the COVID-19 cases of South Korea, China and the United States, we have shown that the proposed estimator outperforms the naive CFR, which provides more accurate estimate during an epidemic and converges faster to the final case fatality rate.

Here is an issue that needs to be stated for clear interpretation of the case fatality rate we estimate. As stated in [23], the proposed estimator and the naive CFR do not estimate the true death rate for the COVID-19 disease, which is defined as the proportion of deaths out of all

## Daily Case Fatality Rate in the United States

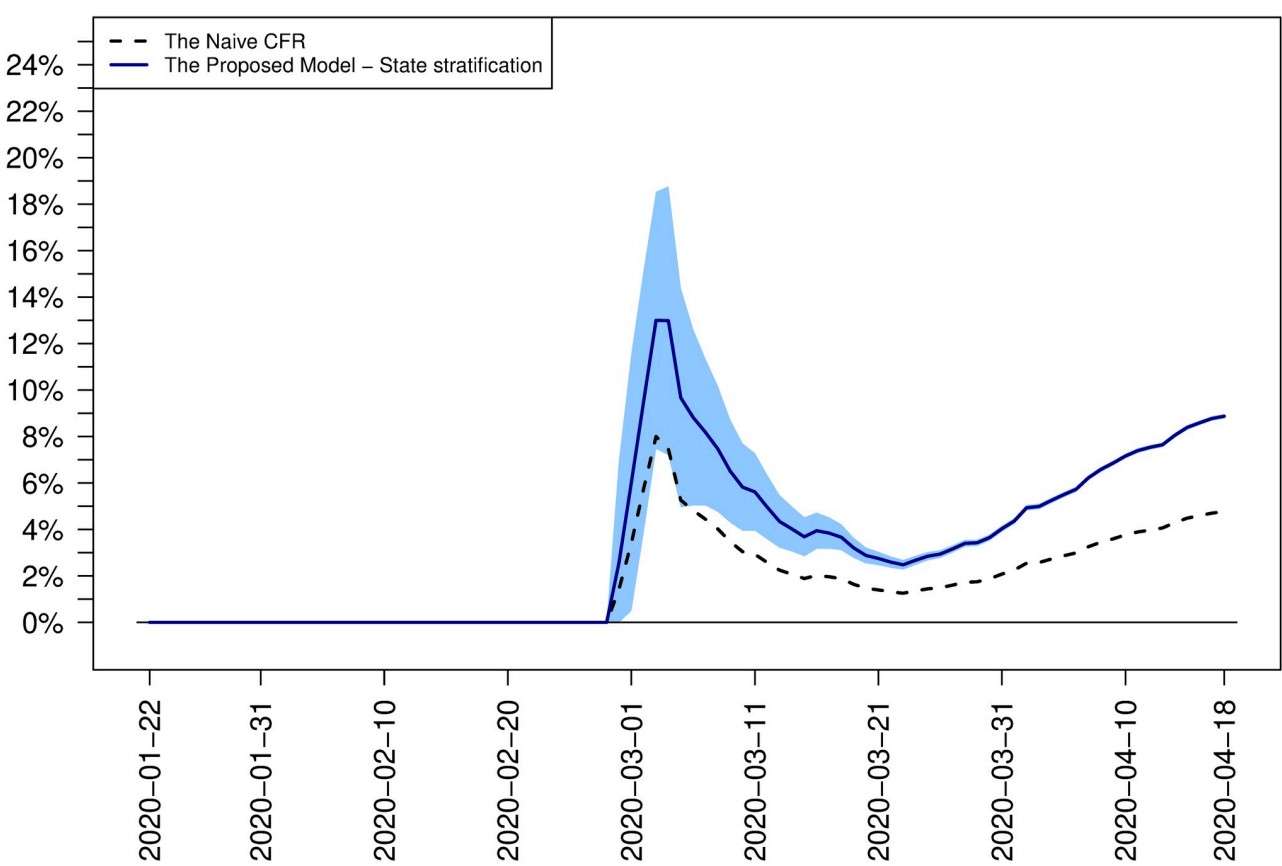

**Fig 5. Daily estimates of CFRs in the United States.** Two CFR estimates in the United States, the daily updated naive CFRs (dotted line) and the proposed estimates of CFRs with state stratification (blue solid line) are depicted together. For our model estimates, daily basis 95% confidence intervals (shaded area) are overlaid.

infected cases. Note that all infected cases include the cases with and without symptoms regardless the disease diagnosed or not. This is the problem caused by not counting on people who are infected the disease but not diagnosed in the estimation of the case fatality rate. Under ordinary circumstances, even an infected person does not take a diagnostic test if there is no or little symptom of the disease. In addition, for the current COVID-19 disease, the shortage of the diagnostic kits and expensive costs required for the test and the cure have prevented from detecting all infected patients (even with symptoms) by making people hesitate to take the diagnostic test.

Even in South Korea, a counter example that a large number of infected cases without symptoms of the disease are included in the cumulative number of confirmed cases, it is not guaranteed that all infected cases are detected. In practice, a larger part of asymptomatic cases cannot be detected unless all members of the population are tested.

Accordingly, the number of confirmed cases is counted to be much smaller than the number of infected cases, and as a result, the estimated case fatality rate tends to be higher than the true death rate. Hence, when the proposed estimator is used for the estimation of the case fatality rate as a measure of severity of a disease, this should be stated. Otherwise, to use the

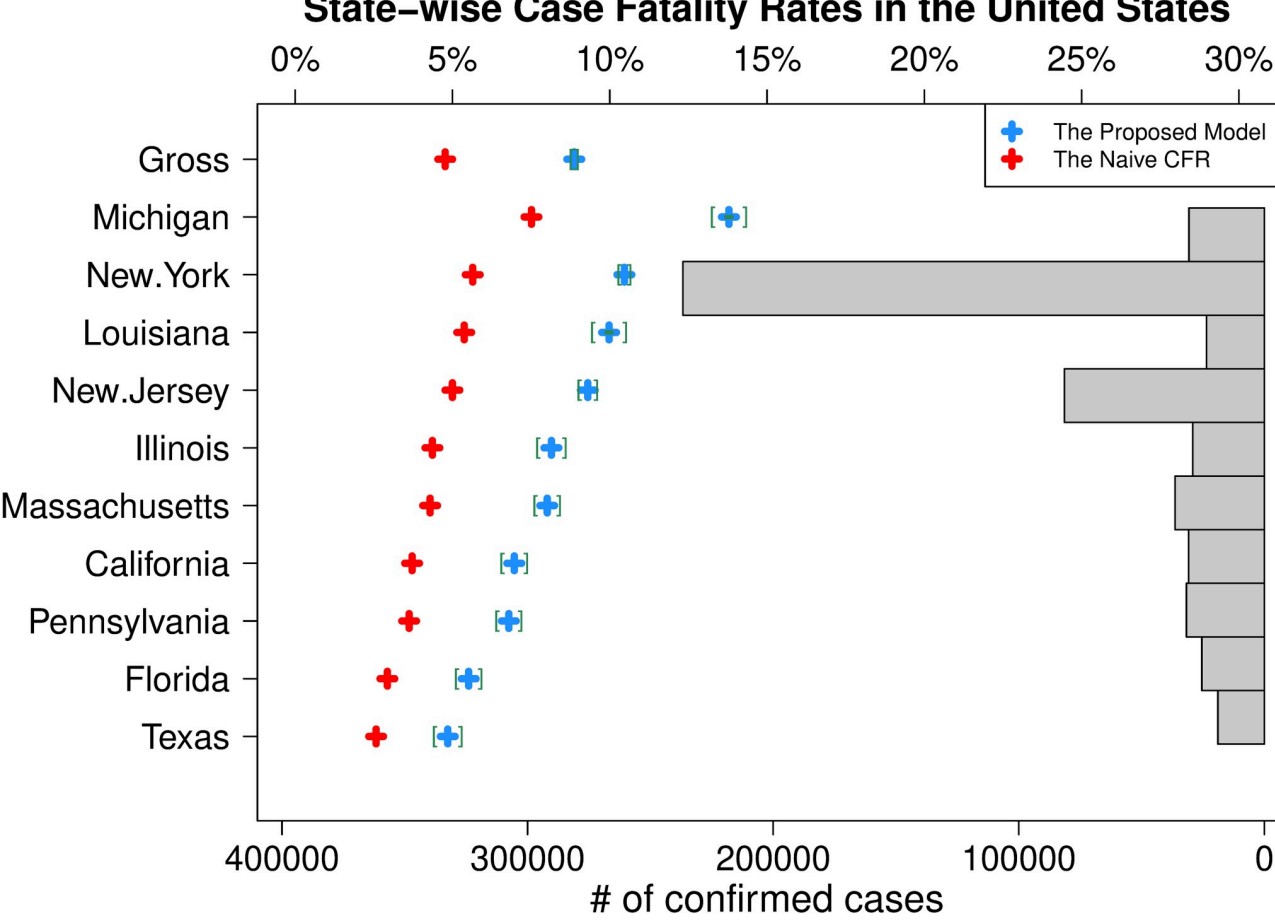

**Fig 6. Latest state-wise CFRs in the United States.** For top 10 states in the numbers of confirmed cases, the latest proposed estimate of CFR (blue plus marker) and its 95% confidence interval (green square brackets) for each state is depicted with the histogram of confirmed cases. The latest naive CFR (red plus marker) calculated for each state separately is also added to show the difference.

proposed estimator as a measure of the true death rate, integration with an accurate estimation of the transmissibility of a disease must be required.

## Supporting information

**S1 Data.**
(TXT)

**S1 Appendix.**
(PDF)

## Author Contributions

**Conceptualization:** Woncheol Jang, Sungkyu Jung, Johan Lim.

**Data curation:** Byungwon Kim, Seonghong Kim.

**Formal analysis:** Byungwon Kim, Seonghong Kim.

**Funding acquisition:** Woncheol Jang, Johan Lim.

**Investigation:** Johan Lim.

**Methodology:** Byungwon Kim, Woncheol Jang, Sungkyu Jung, Johan Lim.

**Supervision:** Woncheol Jang, Sungkyu Jung, Johan Lim.

**Visualization:** Seonghong Kim.

**Writing – original draft:** Byungwon Kim.

**Writing – review & editing:** Woncheol Jang, Sungkyu Jung, Johan Lim.

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
