## [Decision Letter · Decision Letter 0]

9 Jul 2020

PONE-D-20-13762

Estimation of the case fatality rate based on stratification for the COVID-19 outbreak

PLOS ONE

Dear Dr. Jang,

Thank you for submitting your manuscript to PLOS ONE. After careful consideration, we feel that it has merit but does not fully meet PLOS ONE’s publication criteria as it currently stands. Therefore, we invite you to submit a revised version of the manuscript that addresses the points raised during the review process.

We look forward to receiving your revised manuscript.

Kind regards,

Mihye Ahn, Ph.D

Academic Editor

PLOS ONE

Journal Requirements:

Reviewers' comments:

Reviewer's Responses to Questions

**Comments to the Author**

1. Is the manuscript technically sound, and do the data support the conclusions?

Reviewer #1: Partly

Reviewer #2: Yes

Reviewer #3: Yes

2. Has the statistical analysis been performed appropriately and rigorously? 

Reviewer #1: N/A

Reviewer #2: Yes

Reviewer #3: Yes

3. Have the authors made all data underlying the findings in their manuscript fully available?

Reviewer #1: Yes

Reviewer #2: Yes

Reviewer #3: No

4. Is the manuscript presented in an intelligible fashion and written in standard English?

Reviewer #1: Yes

Reviewer #2: Yes

Reviewer #3: Yes

5. Review Comments to the Author

Reviewer #1: The case fatality rate (CFR), a key indicator for a communicable disease, is used to evaluate the disease severity and develop the control policy. The accurate calculation of the CFR is very important for the evaluation of disease control and treatment. The calculation methods of the infectious disease have been well studied in the past, and can be used in a new infectious disease directly.

At present, the CFR of COVID-19 has been studied by many countries, organizes and researchers. There are some impacting factors of the COVID-19 CFR:

1. The CFR is very strongly impacted by the control strategy and hospital resources of the country.

2. When calculating the CFR, we need to know the number of actual cases (not merely the reported ones, which are typically only a small portion of the actual ones). A sampling COVID-19 antibody screening was conducted in New York city, USA. The results showed the actual number was 10 times the number of confirmed cases.

3. The death number also significantly varied in the different countries because of the age of COVID-19 patients.

Above results showed the accurate estimation of the COVID-19 CFR is so complicated and impacted by many factors. So the authors of this manuscript should consider more factors in their model to get closely to the actual CFR.

Reviewer #2: This paper points the underestimation property of crude case fatality rate (CFR), defined by the number of deaths among the patients with a confirmed disease due to the under reported/under collected confirmed cases and proposes an alternative real-time estimator for CFR. This paper addresses a focal problem in estimating time-sensitive quantity, CFR during a disease pandemic. The proposed estimator was applied to COVID-19 data collected in China, South Korea and the United States There are left room in this paper to be elaborated and more tightly written. Major and minor comments can be found in the attached review.

Reviewer #3: The paper develops a new estimation method for the case fatality rate (CFR) of a disease by taking into account potential deaths among confirmed patients alive. The method can overcome the underestimation problem of the naive CFR in an ongoing pandemic. The CFR estimation that the paper deals with is a timely research topic since the proposed method can evaluate the risk of COVID-19, an acute respiratory infectious disease that has triggered a global health crisis. The idea is novel and the paper demonstrates the utility of the proposed method with real COVID-19 data in a proper manner. The paper conducts the estimation of COVID-19 CFRs in China, South Korea, and U.S. Still, the lack of testing creates the overestimation problem since many of confirmed cases are excluded from the data. This practical limit is described in Discussion.

Data availability is not well described for reproducible research. Please add a detailed description in Data Availability Section, so readers can access the data on their own. Some language used can be improved and grammatical errors should be fixed. More details can be found in the attached.

6. PLOS authors have the option to publish the peer review history of their article (what does this mean?). If published, this will include your full peer review and any attached files.

Reviewer #1: No

Reviewer #2: No

Reviewer #3: No

---

## [Author Response · Author response to Decision Letter 0]

5 Aug 2020

See the attached response to reviewers

---

## [Decision Letter · Decision Letter 1]

22 Sep 2020

PONE-D-20-13762R1

Estimation of the case fatality rate based on stratification for the COVID-19 outbreak

PLOS ONE

Dear Dr. Jang,

Thank you for submitting your manuscript to PLOS ONE. After careful consideration, we feel that it has merit but does not fully meet PLOS ONE’s publication criteria as it currently stands. Therefore, we invite you to submit a revised version of the manuscript that addresses the points raised during the review process.

We look forward to receiving your revised manuscript.

Kind regards,

Mihye Ahn, Ph.D

Academic Editor

PLOS ONE

Reviewers' comments:

Reviewer's Responses to Questions

**Comments to the Author**

1. If the authors have adequately addressed your comments raised in a previous round of review and you feel that this manuscript is now acceptable for publication, you may indicate that here to bypass the “Comments to the Author” section, enter your conflict of interest statement in the “Confidential to Editor” section, and submit your "Accept" recommendation.

Reviewer #2: All comments have been addressed

Reviewer #3: (No Response)

2. Is the manuscript technically sound, and do the data support the conclusions?

Reviewer #2: Yes

Reviewer #3: Yes

3. Has the statistical analysis been performed appropriately and rigorously? 

Reviewer #2: Yes

Reviewer #3: Yes

4. Have the authors made all data underlying the findings in their manuscript fully available?

Reviewer #2: Yes

Reviewer #3: Yes

5. Is the manuscript presented in an intelligible fashion and written in standard English?

Reviewer #2: Yes

Reviewer #3: Yes

6. Review Comments to the Author

Reviewer #2: (No Response)

Reviewer #3: I think authors have addressed nearly all the points I raised in the first report. I have a few further comments as a follow-up to some of my previous comments.

7. PLOS authors have the option to publish the peer review history of their article (what does this mean?). If published, this will include your full peer review and any attached files.

Reviewer #2: No

Reviewer #3: No

---

## [Author Response · Author response to Decision Letter 1]

25 Sep 2020

See the attached response letter.

---

## [Decision Letter · Decision Letter 2]

29 Jan 2021

Estimation of the case fatality rate based on stratification for the COVID-19 outbreak

PONE-D-20-13762R2

Dear Dr. Jang,

We’re pleased to inform you that your manuscript has been judged scientifically suitable for publication and will be formally accepted for publication once it meets all outstanding technical requirements.

Kind regards,

Mihye Ahn, Ph.D

Academic Editor

PLOS ONE

Additional Editor Comments (optional):

Reviewers' comments:

Reviewer's Responses to Questions

**Comments to the Author**

1. If the authors have adequately addressed your comments raised in a previous round of review and you feel that this manuscript is now acceptable for publication, you may indicate that here to bypass the “Comments to the Author” section, enter your conflict of interest statement in the “Confidential to Editor” section, and submit your "Accept" recommendation.

Reviewer #2: All comments have been addressed

Reviewer #3: All comments have been addressed

2. Is the manuscript technically sound, and do the data support the conclusions?

Reviewer #2: Yes

Reviewer #3: Yes

3. Has the statistical analysis been performed appropriately and rigorously? 

Reviewer #2: Yes

Reviewer #3: Yes

4. Have the authors made all data underlying the findings in their manuscript fully available?

Reviewer #2: Yes

Reviewer #3: Yes

5. Is the manuscript presented in an intelligible fashion and written in standard English?

Reviewer #2: Yes

Reviewer #3: Yes

6. Review Comments to the Author

Reviewer #2: The revised manuscript reads well. I have no additional comments to improve the readability of the manuscript.

Reviewer #3: (No Response)

7. PLOS authors have the option to publish the peer review history of their article (what does this mean?). If published, this will include your full peer review and any attached files.

Reviewer #2: No

Reviewer #3: No

---

## [Editor Report · Acceptance letter]

5 Feb 2021

PONE-D-20-13762R2 

Estimation of the case fatality rate based on stratification for the COVID-19 outbreak 

Dear Dr. Jang:

I'm pleased to inform you that your manuscript has been deemed suitable for publication in PLOS ONE. Congratulations! Your manuscript is now with our production department. 

Kind regards, 

on behalf of

Dr. Mihye Ahn 

Academic Editor

PLOS ONE